# Learning 3D Anisotropic Noise Distributions Improves Molecular Force Field Modeling

**Xixian Liu**[1][*], **Rui Jiao**[2][*], **Zhiyuan Liu**[3][†], **Yurou Liu**[4], **Yang Liu**[2],
**Ziheng Lu**[5], **Wenbing Huang**[4][†], **Yang Zhang**[3], **Yixin Cao**[6]

[1]Fudan University, [2]Tsinghua University, [3]National University of Singapore
[4]Renmin University of China, [5] Microsoft Research
[6]Institute of Trustworthy Embodied AI, Fudan University
`xixian.liu@mila.quebec, jiaor21@mails.tsinghua.edu.cn`
`zhiyuan@nus.edu.sg, hwenbing@ruc.edu.cn`

## Abstract

Coordinate denoising has emerged as a promising method for 3D molecular pre-training due to its theoretical connection to learning a molecular force field. However, existing denoising methods rely on oversimplified molecular dynamics that assume atomic motions to be isotropic and homoscedastic. To address these limitations, we propose a novel denoising framework **AniDS**: Anisotropic Variational Autoencoder for 3D Molecular Denoising. AniDS introduces a structure-aware anisotropic noise generator that can produce atom-specific, full covariance matrices for Gaussian noise distributions to better reflect directional and structural variability in molecular systems. These covariances are derived from pairwise atomic interactions as anisotropic corrections to an isotropic base. Our design ensures that the resulting covariance matrices are symmetric, positive semi-definite, and SO(3)-equivariant, while providing greater capacity to model complex molecular dynamics. Extensive experiments show that AniDS outperforms prior isotropic and homoscedastic denoising models and other leading methods on the MD17 and OC22 benchmarks, achieving average relative improvements of **8.9%** and **6.2%** in force prediction accuracy. Our case study on a crystal and molecule structure shows that AniDS adaptively suppresses noise along the bonding direction, consistent with physicochemical principles. Our code is available at https://github.com/ZeroKnighting/AniDS.

## 1 Introduction

Accurately and efficiently predicting atomic properties of molecules and materials is fundamental to a wide range of downstream applications, including drug discovery [1, 2], material design [3], catalyst design [4, 5], and molecular dynamics simulations [6, 7]. *Ab initio* methods such as Density Functional Theory (DFT) [8] are widely regarded as the standard for atomic property prediction, but their high computational cost significantly limits scalability to large systems or datasets. To overcome this challenge, machine learning potentials [9] have been developed to accelerate atomic simulations by orders of magnitude, with recent advances of graph neural networks (GNNs) [10, 11] further improving prediction accuracy. Among these methods, coordinate denoising [12, 13] has emerged as a promising training objective, due to its theoretical grounding in learning molecular force fields.

Classical coordinate denoising [12] involves recovering the original atomic coordinates $\mathbf{X}$ from the perturbed version $\mathbf{X} + \boldsymbol{\epsilon}$, where the noise $\boldsymbol{\epsilon} \sim \mathcal{N}(\mathbf{0}, \sigma^2 \mathbf{I})$ is drawn from an isotropic Gaussian. This objective corresponds to learning the molecular force fields when the data distribution $p(\tilde{\mathbf{X}})$

---

[*]Equal contribution. † Correspondence.

39th Conference on Neural Information Processing Systems (NeurIPS 2025).

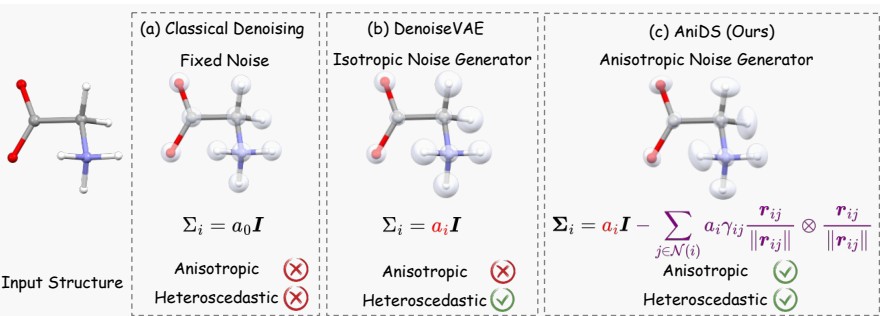

Figure 1: Comparison between different denoising approaches. The opaque spheres represent noise distributions. Our approach captures noise distribution that is both anisotropic and heteroscedastic.

is approximated as a mixture of isotropic Gaussians centered at each data point [13], *i.e.,* $p(\tilde{\mathbf{X}}) \approx \frac{1}{n} \sum_{i=1}^{n} \mathcal{N}(\tilde{\mathbf{X}}|\mathbf{X}_i, \sigma^2\mathbf{I})$. A closer examination of this formulation reveals two implicit assumptions: (1) atomic motions are assumed to be **isotropic**, with identical variance along all axes ($\sigma_x^2 = \sigma_y^2 = \sigma_z^2$), ignoring direction-dependent stiffness such as bond stretching [14]; and (2) atomic motions are assumed to be **homoscedastic** [15], with all atoms sharing the same noise scale $\sigma^2$, ignoring variability in energy potentials across distinct structures. While these assumptions make the implementation easier, they oversimplify the molecular dynamics, potentially leading to inaccurate force field learning.

Relaxing these assumptions demands a new noise distribution that is theoretically consistent with the assumed atomic motion. DenoiseVAE [16] learns adaptive noise scale $\sigma^2$ for different atoms, lifting the assumption of homoscedasticity, although the isotropic assumption remains less explored. On the other hand, earlier works [17, 18] move beyond isotropic Gaussian noises by introducing handcrafted noises, *e.g.,* perturbing bonds and dihedrals. However, these noises are applied independently at fixed scales, thus failing to address the assumption of homoscedasticity.

In this work, we propose a novel denoising framework: **Anisotropic Variational Autoencoder for 3D Molecular Denoising** (**AniDS**), aiming to improve molecular force field learning by adaptively generating anisotropic Gaussian noises for coordinate denoising, thereby lifting all three assumptions discussed above. AniDS achieves this by adaptively generating a "full covariance matrix" (*i.e.,* without the diagonal constraint) for each atom's Gaussian noise distribution, enabling it to capture directional variability and different molecular structures. However, generating such full covariance matrix introduces two major challenges that have hindered prior efforts: (1) Ensuring the covariance matrices are symmetric, positive semi-definite, and equivariant, which are fundamental properies of covariance matrices and molecules. While these properties are trivially satisfied in isotropic settings, they require explicit modeling and constraint in the anisotropic case. (2) Guiding the covariance learning process with proper physicochemical priors. The space of full covariance matrices is high-dimensional and susceptible to trivial or degenerated solutions if not properly regularized, which often leads to unstable or failed pretraining.

To address these challenges, AniDS introduces a **structure-aware anisotropic noise generator** that leverages 3D molecular structures to produce atom-specific full covariance matrices for anisotropic noise distributions. Specifically, the generator first constructs an isotropic base term and then applies anisotropic corrections derived from the learned pairwise atomic interactions. These anisotropic corrections enable the model to capture local rigidity and directional variability across different molecular structures, acting as a physicochemical prior. Additionally, our design ensures the covariance matrices are symmetric, positive semi-definite, and SO(3)-equivariant. Equipped with this expressive noise generator, AniDS supervises a molecular denoising autoencoder using our theoretically grounded denoising loss that approximates learning the molecular force field.

AniDS achieves an average relative improvement of **8.9%** on MD17 and **6.2%** on OC22 in force prediction. Ablation studies confirm the effectiveness of each component. Our case study further shows that AniDS adaptively adjusts the direction and scale of the noise according to the atomic interactions' strength in $H_3In_{12}O_{48}Pd_{12}$ and $SNPH_4$ structures, aligning with physicochemical principles.

## 2 Preliminary: Coordinate Denoising and DenoiseVAE

We begin by introducing the coordinate denoising [13] objective and the subsequent study DenoiseVAE [16] that addresses the homoscedastic assumption.

**Notation of Molecules.** In this work, we develop a framework to learn force fields for both small molecules and crystal materials. For clarity, we present the methodology using the notation of small molecules; the extension to crystals is straightforward and involves revising the denoising autoencoder to reflect crystals' periodicity. Details of this adaptation are provided in Appendix B.

A 3D molecule $\mathbf{M} = (\mathbf{Z}, \mathbf{X}, \mathbf{E})$ is represented by: (1) $\mathbf{Z} \in \mathbb{N}^N$, with $\mathbf{Z}_i \in \mathbb{N}$ denotes the atomic number of the $i$-th atom; (2) $\mathbf{X} \in \mathbb{R}^{N \times 3}$, where $\mathbf{X}_i \in \mathbb{R}^3$ specifies the 3D coordinates of the $i$-th atom; and (3) $\mathbf{E} \in \mathbb{R}^{N \times N \times d}$, where $\mathbf{E}_{ij} \in \mathbb{R}^d$ denotes the bond existence and bond type between atoms $i$ and $j$. Here, $N$ denotes the number of atoms in the molecule.

**Coordinate Denoising.** Given a molecule $\mathbf{M}$ in equilibrium, in which the atom-wise forces are near zero, a corrupted version is generated as $\tilde{\mathbf{M}} = (\mathbf{Z}, \tilde{\mathbf{X}}, \mathbf{O})$, where the atomic coordinates are perturbed by Gaussian noise: $\tilde{\mathbf{X}} = \mathbf{X} + \sigma \boldsymbol{\epsilon}$ with $\boldsymbol{\epsilon} \sim \mathcal{N}(\mathbf{0}, \mathbf{I})$. Here $\sigma$ is a hyperparameter controlling the added noise scale, which is usually around $0.1$ to generate a small perturbation. Then, a denoising autoencoder [19–21] $\phi(\cdot)$ is trained to predict the added noise by minimizing the following loss: $\mathbb{E}_{p(\tilde{\mathbf{M}}, \mathbf{M})} \| \phi(\tilde{\mathbf{M}}) - (\tilde{\mathbf{X}} - \mathbf{X}) / \sigma^2 \|^2$. Theoretically, this denoising objective is shown to approximate learning a molecular force field [13], therefore improving its performance.

**DenoiseVAE [16].** The standard coordinate denoising operates under the homoscedasticity assumption, treating all atoms equally. Specifically, a fixed noise scale $\sigma$ is used to simulate minor thermal fluctuations for all atoms, but fails to capture the energy variations across different molecular structures. For example, in a rigid structure like a benzene ring, a minor coordinate perturbation can lead to a disproportionately large energy change, whereas similar perturbations in more flexible regions may have negligible energetic impact. Resolving this issue, DenoiseVAE trains a noise generator $\psi(\cdot)$ to adaptively assign a distinct noise scale $\sigma_i$ to each atom based on the molecular structure:

$$\{\sigma_i \in \mathbb{R}^+ \mid i \in \mathbf{M}\} = \psi(\mathbf{M}), \quad (1) \qquad \tilde{\mathbf{X}}_i = \mathbf{X}_i + \sigma_i \boldsymbol{\epsilon}_i, \qquad \boldsymbol{\epsilon}_i \sim \mathcal{N}(\mathbf{0}, \mathbf{I}). \quad (2)$$

We then obtain the perturbed molecule $\tilde{\mathbf{M}} = (\mathbf{Z}, \tilde{\mathbf{X}}, \mathbf{L})$, and perform denoise training as follows:

$$\mathcal{L}_{\text{Denoise}} = \frac{1}{|\mathbf{M}|} \mathbb{E}_{p(\tilde{\mathbf{X}}, \mathbf{X})} \sum_{i \in \mathbf{M}} \sigma_i^2 \| \phi(\tilde{\mathbf{M}}) - \frac{(\tilde{\mathbf{X}} - \mathbf{X})}{\sigma_i^2} \|^2, \tag{3}$$

$$\mathcal{L}_{\text{KL}} = \frac{1}{|\mathbf{M}|} \sum_{i \in \mathbf{M}} \mathbf{D}_{\text{KL}}(\mathcal{N}(\mathbf{0}, \sigma_i^2 \mathbf{I}) \| p_i), \tag{4}$$

where $p_i$ is the prior distribution, set to $\mathcal{N}(\mathbf{0}, \sigma_p^2 \mathbf{I})$, and $\sigma_p$ is a hyperparameter. The KL divergence term $\mathcal{L}_{\text{KL}}$ acts as a regularizer to prevent the noise generator $\psi(\cdot)$ from collapsing to trivial solutions where $\sigma_i \to 0$. This ensures meaningful denoise training. The final training objective combines both losses $\mathcal{L}_{\text{DenoiseVAE}} = \lambda_{\text{Denoise}} \mathcal{L}_{\text{Denoise}} + \lambda_{\text{KL}} \mathcal{L}_{\text{KL}}$ with balancing coefficients $\lambda_{\text{Denoise}}$ and $\lambda_{\text{KL}}$.

## 3 Methodology

In this section, we start by overviewing the AniDS framework. We then present the motivations and detailed formulations of its key components in Sections 3.2 and Section 3.3, respectively. Finally, we describe how AniDS can be adapted to different training schemes in Section 3.4.

### 3.1 Overview of the AniDS Framework

AniDS consists of three key components: (1) a structure-aware anisotropic noise generator $\psi(\cdot)$ that generates per-atom noise distribution conditioned on the molecule structure $\mathbf{M}$; (2) a denoising autoencoder $\phi(\cdot)$ that is trained for coordinate denoising; and (3) a denoising objective $\mathcal{L}_{\text{AniDS}}$ that approximates learning molecular force field.

**Structure-aware Anisotropic Noise Generator.** Given a molecule $\mathbf{M}$, AniDS employs a noise generator $\psi(\cdot)$ to generate a full covariance matrix $\boldsymbol{\Sigma}_i \in \mathbb{R}^{3 \times 3}$ for each atom $i \in \mathbf{M}$, defining an anisotropic Gaussian noise distribution $\mathcal{N}(\mathbf{0}, \boldsymbol{\Sigma}_i)$. Unlike prior works that assume diagonal $\boldsymbol{\Sigma}_i$ for theoretical and computational convenience [22, 13], AniDS models $\boldsymbol{\Sigma}_i$ as a dense matrix to capture directional variability. To ensure that the generated noise reflects the underlying molecular structure, $\psi(\cdot)$ should be implemented with structure-aware molecular encoders [23, 24]. This enables the generated covariance matrices to reflect local and global molecular structural contexts. Given this generator $\psi(\cdot)$, the perturbed molecule $\tilde{\mathbf{M}}$ can be obtained as:

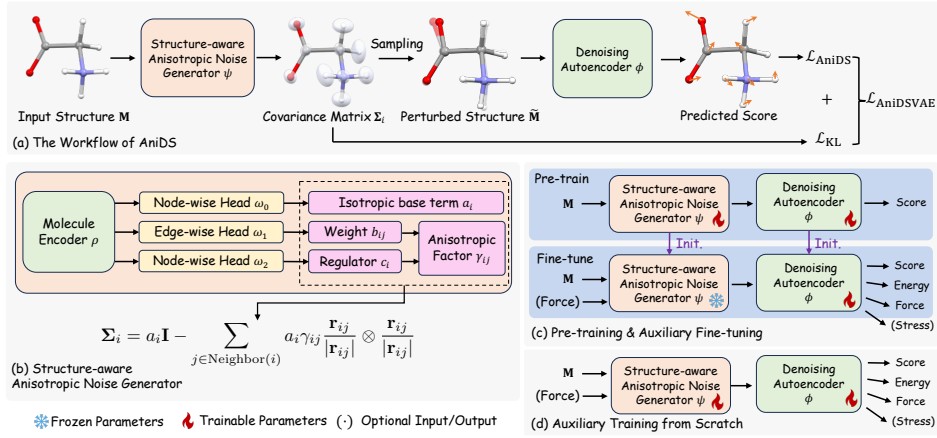

Figure 2: Overview of the AniDS framework.

$$\{\mathbf{\Sigma}_i \in \mathbb{R}^{3\times 3} | i \in \mathbf{M}\} = \psi(\mathbf{M}), \quad (5) \qquad \mathbf{L}_i = \text{Cholesky}(\mathbf{\Sigma}_i), \quad (6)$$

$$\tilde{\mathbf{X}}_i = \mathbf{X}_i + \mathbf{L}_i\boldsymbol{\epsilon}_i, \boldsymbol{\epsilon}_i \sim \mathcal{N}(\mathbf{0}, \mathbf{I}), \quad (7) \qquad \tilde{\mathbf{M}} = (\mathbf{Z}, \tilde{\mathbf{X}}, \mathbf{E}), \quad (8)$$

where $\tilde{\mathbf{X}}_i$ is the perturbed coordinate of the $i$-th atom, sampled via Cholesky decomposition [25] of the learned covariance $\mathbf{\Sigma}_i$. While the noise genrator can be any neural network capable of producing symmetric, positive semi-definite, and 3D-equivariant covariance matrices, we introduce our implementation in Section 3.2.

**Denoising Autoencoder.** The denoising autoencoder $\phi$ takes the perturbed structure $\tilde{\mathbf{M}}$ as input and predicts atom-wise vectors intended to recover the original coordinates: $\phi(\tilde{\mathbf{M}}) \in \mathbb{R}^{N\times 3}$. AniDS is a model-agnostic framework that can be integrated into a wide range of molecular backbones. In this work, we implement $\phi$ with EquiformerV2 [24] and Geoformer [26], which have demonstrated strong performance on molecular benchmarks. To enable coordinate denoising on non-equilibrium structures, we follow [27] to include **force encoding** as the additional features for the corrupted atoms. Unlike equilibrium structures, which are uniquely defined by local energy minima, multiple non-equilibrium structures may share the same energy level, introducing ambiguity into the denoising targets. Including force information helps disambiguate these cases by anchoring the target to a specific molecular state. Our detailed implementations of the denoising autoencoders and the force encoding are introduced in Appendix B.1.

**AniDS's Denoising Objective.** AniDS's denoising loss jointly trains the noise generator and the denoising autoencoder to recover the noise vector scaled by the inverse covariance:

$$\mathcal{L}_{\text{AniDS}} = \frac{1}{|\mathbf{M}|} \mathbb{E}_{q_{\mathbf{\Sigma}}(\tilde{\mathbf{X}}, \mathbf{X})} \sum_{i\in\mathbf{M}} \left\| \phi(\tilde{\mathbf{M}})_i - [\mathbf{\Sigma}_i]^{-1}(\tilde{\mathbf{X}}_i - \mathbf{X}_i) \right\|^2 . \quad (9)$$

We show in Section 3.3 that the objective above approximates learning the molecular force field. Additionally, we compute a KL divergence loss between the learned anisotropic noise distribution and an isotropic Gaussian prior $\mathcal{N}(0, \sigma_p^2 I)$:

$$\mathcal{L}_{\text{KL}} = \frac{1}{2|\mathbf{M}|} \sum_{i\in\mathbf{M}} \left( \text{tr}(\sigma_p^{-2}\mathbf{\Sigma}_i) - d + \ln\frac{|\sigma_p^2 I|}{|\mathbf{\Sigma}_i|} \right), \quad (10)$$

which prevents the learned covariance from collapsing into trivial solutions like zeros. Recall that the normalized anisotropic weights $\{\gamma_{ij}\}$ on Eq. 15 satisfy $\sum_j \gamma_{ij} < 1$ by construction. To avoid degenerate covariances that become overly isotropic in flexible regions, we regularize the total anisotropic mass $\Gamma_i := \sum_j \gamma_{ij}$ towards a target level $\kappa \in (0, 1)$ via a one-sided hinge-squared penalty:

$$\mathcal{L}_\gamma = \frac{1}{|\mathcal{M}|} \sum_{i\in\mathcal{M}} \left[ \max(0, \kappa - \Gamma_i) \right]^2 . \quad (11)$$

Intuitively, Eq. (11) discourages vanishing anisotropic corrections (i.e., $\Gamma_i$ too small), while preserving the PSD guarantee $\Gamma_i < 1$ from 15. The final loss $\mathcal{L}_{\text{AniDSVAE}}$ is a weighted sum of the denoising and regularization terms:

$$\mathcal{L}_{\text{AniDS-VAE}} = \lambda_{\text{AniDS}}\mathcal{L}_{\text{AniDS}} + \lambda_{\text{KL}}\mathcal{L}_{\text{KL}} + \lambda_\gamma \mathcal{L}_\gamma . \quad (12)$$

## 3.2 Structure-Aware Anistropic Noise Generation

The noise generator $\psi(\cdot)$ constructs symmetric, positive semi-definite, and SO(3)-equivariant covariance matrices $\{\boldsymbol{\Sigma}_i | i \in \mathbf{M}\}$. These matrices encode both isotropic and anisotropic uncertainty in atomic positions, reflecting each atom's structural context. The generation consists of two main steps:

**Atom-wise Structural Representation.** A molecular structure encoder $\rho(\cdot)$ is used to extract structural representations, to guide the covariance generation process. This work shares the architecture as the denoising autoencoder. Formally, we have:

$$\{\mathbf{h}_i \in \mathbb{R}^d | i \in \mathbf{M}\} = \rho(\mathbf{M}), \tag{13}$$

where $\mathbf{h}_i$ encodes the structural and chemical contexts of atom $i$.

**Covariance Matrix.** Given the structural representation above, we parameterize the covariance matrix $\boldsymbol{\Sigma}_i$ for atom $i$ as the sum of a isotropic base term and an anisotropic term, motivated by our physicochemical prior of pairwise atomic interactions:

$$\boldsymbol{\Sigma}_i = \underbrace{a_i \mathbf{I}}_{\text{Isotropic Base}} - \underbrace{\sum_{j \in \text{Neighbor}(i)} a_i \gamma_{ij} \frac{\mathbf{r}_{ij}}{|\mathbf{r}_{ij}|} \otimes \frac{\mathbf{r}_{ij}}{|\mathbf{r}_{ij}|}}_{\text{Anisotropic Corrections}}, \tag{14}$$

$$\gamma_{ij} = \frac{\exp(b_{ij})}{\sum_{l \in \text{Neighbor}(i)} \exp(b_{il}) + c_i} \quad \text{(Normalized anisotropic weight)}, \tag{15}$$

where $\mathbf{r}_{ij} = \mathbf{X}_i - \mathbf{X}_j \in \mathbb{R}^3$ is the relative position vector. The parameters $a_i$, $b_{ij}$, and $c_i$ are adaptively derived from the atomic structual representations $\{\mathbf{h}_i | i \in \mathbf{M}\}$ using MLPs (*i.e.,* $\omega_0$, $\omega_1$, and $\omega_2$) and a radial basis function [28] $\eta(\cdot)$, respectively:

- **Isotropic base term** $a_i = \exp(\omega_0(\mathbf{h}_i)) \in \mathbb{R}^+$. It controls the baseline noise level for atom $i$, reflecting its intrinsic flexibility. Larger $a_i$ values indicate greater isotropic flexibility.

- **Anisotropic weight** $b_{ij} = \omega_1([\mathbf{h}_i; \mathbf{h}_j; \eta(|\mathbf{r}_{ij}|)]) \in \mathbb{R}$. It controls the weights of anisotropic corrections, which determines the directional stiffness between atoms $i$ and $j$. For example, along covalent bonds, high $b_{ij}$ produces high $\gamma_{ij}$ that suppresses motion along $\mathbf{r}_{ij}$. Conversely, $b_{ij} \ll 0$ for distant atom pairs eliminates the corresponding directional influence of stiffness.

- **Anisotropic regulator** $c_i = \exp(\omega_2(\mathbf{h}_i)) \in \mathbb{R}^+$. It regulates the influence of anisotropic corrections. In symmetric or rigid environments, such as aromatic rings, larger $c_i$ values suppress individual directional components, resulting in a more isotropic noise distribution. In contrast, smaller $c_i$ allows stronger anisotropic corrections to reflect local uncertainty in flexible regions.

Our noise generator is designed to ensure three essential properties of the covariance matrix:

- **Symmetry:** $\boldsymbol{\Sigma}_i = \boldsymbol{\Sigma}_i^\top$ is satisfied due to the symmetry of the isotropic base and the outer product.

- **Positive Semi-Definiteness**: The covariance matrix is positive definite ($\boldsymbol{\Sigma}_i \succ 0$) if $\sum_j \gamma_{ij} < 1$, enforced via the softmax in Equation (15). This condition guarantees the anisotropic correction term does not overpower the isotropic base $a_i \mathbf{I}$. A detailed proof is in Appendix B.

- **SO(3)-Equivariane and T(3)-Invariance**: Given SE(3) transformation $g = (\mathbf{R}, \mathbf{t})$ of rotation $\mathbf{R} \in \text{SO}(3)$ and translation $\mathbf{t} \in \mathbb{R}^3$ applied on the molecule $\mathbf{M}$, the transformed covariance matrix satisfies $g \circ \boldsymbol{\Sigma}_i = \mathbf{R}\boldsymbol{\Sigma}_i\mathbf{R}^\top$: the covariance matrix is equivariant to the rotation but invariant to the translation. This is because the translation is canceled out when computing $\mathbf{r}_{ij} = \mathbf{X}_i - \mathbf{X}_j$.

## 3.3 AniDS Learns Molecular Force Fields

Here, we show that AniDS's denoising objective approximates learning molecular force field.

**Step 1: Boltzmann Distribution and Gaussian Mixture Approximation.** The distribution of molecular structures follows the Boltzmann distribution: $p_{\text{physical}}(\tilde{\mathbf{X}}) \propto \exp(-\frac{E(\tilde{\mathbf{X}})}{k_B T})$, where $E(\tilde{\mathbf{X}})$ is the potential energy. Following prior works [13, 29], we approximate $p_{\text{physical}}$ as a mixture of Gaussians centered at the known structures, typically selected as the equilibrium conformations [13]. Subsequent study [17] demonstrates that learning this Gaussian mixture over non-equilibrium structures is theoretically equivalent to modeling a hybrid noise distribution over equilibrium ones (See

Proposition 3.4 in [17]). In their work, non-equilibrium structures are generated by applying small torsional perturbations to equilibrium conformations. Further studies show that intermediate states from molecular dynamics simulations can also serve as effective non-equilibrium samples for denoising [22, 27], which are also adopted in our work. Specfically, we have:

$$p_{\text{physical}}(\tilde{\mathbf{X}}) \approx q_{\Sigma}(\tilde{\mathbf{X}}) = \frac{1}{|\mathcal{D}|} \sum_{k=1}^{|\mathcal{D}|} q_{\Sigma}(\tilde{\mathbf{X}}|\mathbf{X}^{(k)}), \tag{16}$$

where $\mathbf{X}^{(k)}$ is the $k$-th structure in dataset $\mathcal{D}$. We define $q_{\Sigma}(\tilde{\mathbf{X}}|\mathbf{X}^{(k)}) = \Pi_{i=1}^{N} \mathcal{N}(\tilde{\mathbf{X}}_i; \mathbf{X}_i^{(k)}, \mathbf{\Sigma}_i^{(k)})$. Here $\mathbf{X}_i^{(k)} \in \mathbb{R}^3$ denotes the coordinate of the $i$-th atom in the $k$-th structure, and $\mathbf{\Sigma}_i^{(k)} \in \mathbb{R}^{3\times3}$ is the covariance matrix of the $i$-th atom in the $k$-th structure.

**Step 2: Score Function of the Gaussian Mixture.** The score (gradient of the log-density) is:

$$\nabla_{\tilde{\mathbf{X}}} \log q_{\mathbf{\Sigma}}(\tilde{\mathbf{X}}) = \frac{\sum_{k=1}^{|\mathcal{D}|} q_{\mathbf{\Sigma}}(\tilde{\mathbf{X}}|\mathbf{X}^{(k)}) \nabla_{\tilde{\mathbf{X}}} \log q_{\mathbf{\Sigma}}(\tilde{\mathbf{X}}|\mathbf{X}^{(k)})}{\sum_{k=1}^{|\mathcal{D}|} q_{\mathbf{\Sigma}}(\tilde{\mathbf{X}}|\mathbf{X}^{(k)})} \tag{17}$$

**Step 3: Link to Molecular Force Field.** Under the Boltzmann distribution, the force field is proportional to the score:

$$\mathbf{F}(\tilde{\mathbf{X}}) = -\nabla_{\tilde{\mathbf{X}}} E(\tilde{\mathbf{X}}) = k_B T \cdot \nabla_{\tilde{\mathbf{X}}} \log p_{\text{physical}}(\tilde{\mathbf{X}}). \tag{18}$$

For small perturbation ($\tilde{\mathbf{X}} \approx \mathbf{X}^{(k)}$), the Gaussian mixture is dominated by the nearest structure $\mathbf{X}^{(k)}$, simplifying the score (*cf.* Equation (17)) to:

$$\nabla_{\tilde{\mathbf{X}}} \log q_{\mathbf{\Sigma}}(\tilde{\mathbf{X}}) \approx \nabla_{\tilde{\mathbf{X}}} \log q_{\mathbf{\Sigma}}(\tilde{\mathbf{X}}|\mathbf{X}^{(k)}) = -\sum_{j=1}^{N} [\Sigma_i^{(k)}]^{-1}(\tilde{\mathbf{X}}_i - \mathbf{X}_i^{(k)}). \tag{19}$$

Substituting this into the force field expression (*cf.* Equation (18)), we identify:

$$\mathbf{F}(\tilde{\mathbf{X}}) \propto \sum_{i=1}^{N} [\mathbf{\Sigma}_i^{(k)}]^{-1}(\tilde{\mathbf{X}}_i - \mathbf{X}_i^{(k)}). \tag{20}$$

**Step 4: Denoising as Learning Force Field.** AniDS trains a denoise autoencoder $\phi(\tilde{\mathbf{M}})$ to predict the noise scaled by the inverse covariance (*cf.* Equation (9)). By Vincent's theorem [30], this is equivalent to score matching:

$$\mathcal{L}_{\text{AniDS}} \propto \mathbb{E}_{q_{\mathbf{\Sigma}}(\tilde{\mathbf{X}})} \|\phi(\tilde{\mathbf{M}}) - \nabla_{\tilde{\mathbf{X}}} \log q_{\mathbf{\Sigma}}(\tilde{\mathbf{X}})\|^2 \tag{21}$$

At convergence, $\phi^*(\tilde{\mathbf{M}}) = \nabla_{\tilde{\mathbf{X}}} \log q_{\mathbf{\Sigma}}(\tilde{\mathbf{X}}) \approx \nabla_{\tilde{\mathbf{X}}} \log p_{\text{physical}}(\tilde{\mathbf{X}})$, recovering the force field: $\phi^*(\tilde{\mathbf{M}}) \propto -\mathbf{F}(\tilde{\mathbf{X}})$. Prior works of Coordinate Denoising [27] and DenoiseVAE [16] can be seen as special cases of our approach, with detailed derivations provided in Appendix B.

### 3.4 Adapting AniDS to Different Training Schemes

Consistent with prior works, we adopt AniDS under two training schemes: (1) as a pretraining objective followed by task-specific fine-tuning [13, 31], and (2) as an auxiliary task jointly optimized with supervised force field learning when training from scratch [27, 12].

**Pre-training and Fine-tuning.** Following [12, 31], we can apply AniDS (*cf.* Equation (12)) to pretrain the denoising autoencoder $\phi$ and the structure-aware noise generator $\psi$ altogether on a large pretraining dataset. The pretrained encoder $\phi$ is then fine-tuned on downstream datasets for supervised force field learning, with the AniDS objective retained as an auxiliary loss, as described in the following paragraph. During fine-tuning, the parameters of the noise generator $\psi$ are kept frozen.

**Supervised Learning with Partial Corruption and Auxiliary Denoising.** We use AniDS as an auxiliary loss for supervised force field learning. Inspired by [27], we adopt a partially corrupted denoising strategy. Specifically, only a subset of a molecule's atom coordinates is corrupted using the noise generator. The model is trained with the weighted sum of supervised force field loss on the uncorrupted atoms and AniDS loss (*cf.* Equation (12)) on the corrupted atoms. This design ensures that the model learns from clean ground truth coordinates for supervised force field learning, while still benefiting from denoising regularization. Compared to corrupting all the atoms, this method mitigates the mismatch between the the ground truth force field label and the perturbed structure. Implementation details are provided in Appendix B.3.

Table 1: MAE for force prerdiction on MD17's test sets. Forces are reported in units of kcal/mol. Bold numbers indicate the best performance. Green denotes relative improvement to the baseline.

| Model | Aspirin | Benzene | Ethanol | Malonaldehyde | Naphthalene | Salicylic Acid | Toluene | Uracil | Avg |
|---|---|---|---|---|---|---|---|---|---|
| SchNet [37] | 1.35 | 0.31 | 0.39 | 0.66 | 0.58 | 0.85 | 0.57 | 0.56 | 0.66 |
| DimeNet [38] | 0.499 | 0.187 | 0.230 | 0.383 | 0.215 | 0.374 | 0.216 | 0.301 | 0.300 |
| PaiNN [39] | 0.338 | - | 0.224 | 0.319 | 0.077 | 0.195 | 0.094 | 0.139 | - |
| TorchMD-NET [40] | 0.253 | 0.196 | 0.109 | 0.169 | 0.061 | 0.129 | 0.067 | 0.095 | 0.135 |
| NequIP (Lmax=3) [10] | 0.184 | - | 0.071 | 0.129 | 0.039 | 0.090 | 0.046 | 0.076 | - |
| SE(3)-DDM [41] | 0.453 | - | 0.166 | 0.288 | 0.129 | 0.266 | 0.122 | 0.122 | - |
| Coord [13] | 0.211 | 0.169 | 0.096 | 0.139 | 0.053 | 0.109 | 0.058 | 0.074 | 0.114 |
| Frad [17] | 0.209 | 0.199 | 0.091 | 0.1415 | 0.053 | 0.108 | 0.054 | 0.076 | 0.116 |
| Slide [18] | 0.174 | 0.169 | 0.088 | 0.153 | 0.048 | 0.100 | 0.054 | 0.082 | 0.109 |
| DeNS [27] (Lmax=2) | 0.131 | 0.141 | 0.060 | 0.101 | 0.039 | 0.085 | 0.044 | 0.076 | 0.085 |
| DeNS [27] (Lmax=3) | 0.120 | 0.141 | 0.055 | 0.095 | 0.037 | 0.074 | 0.042 | 0.067 | 0.079 |
| AniDS (Lmax=2) | **0.102**$^{+15.0\%}$ | **0.139**$^{+1.4\%}$ | **0.050**$^{+9.1\%}$ | **0.084**$^{+11.6\%}$ | **0.036**$^{+2.7\%}$ | **0.064**$^{+13.5\%}$ | **0.038**$^{+9.5\%}$ | **0.062**$^{+7.5\%}$ | **0.072**$^{+8.9\%}$ |

Table 2: Performance comparison on OC22's S2EF-Total validation set.

| Model | Params | Energy E-MAE (meV) | | | Force F-MAE (meV/Å) | | |
|---|---|---|---|---|---|---|---|
| | | ID | OOD | Avg | ID | OOD | Avg |
| GemNet-OC [42] | 39M | 545 | 1011 | 778 | 30.0 | 40.0 | 35.0 |
| GemNet-OC (OC20+OC22) [42] | 39M | 464 | 859 | 661.5 | 27.0 | 34.0 | 30.5 |
| E2former [43] | 67M | 491 | 724 | 607.5 | 25.98 | 36.45 | 31.22 |
| EquiformerV2 [24] | 122M | 433.0 | 629.0 | 531 | 22.88 | 30.70 | 26.79 |
| EquiformerV2 + DeNS [27] | 127M | 391.6 | 533.0 | 462.3 | 20.66 | 27.11 | 23.89 |
| EquiformerV2 + AniDS (ours) | 129M | **370.0**$^{+5.5\%}$ | **525.4**$^{+1.4\%}$ | **447.7**$^{+3.2\%}$ | **19.53**$^{+5.5\%}$ | **25.27**$^{+6.8\%}$ | **22.4**$^{+6.2\%}$ |

## 4 Experiments

**Datasets.** We briefly describe the datasets in our experiments, and leave additional setup details in Appendix C.2. Our experiments involve four datasets: (1) **PCQM4Mv2** [32] contains 3,746,619 molecules along with their Density Functional Theory (DFT) calculated 3D equilibrium structures. (2) **OC22** [33] is a large-scale dataset of DFT calculated structures designed to advance machine learning for oxide electrocatalysts. (3) **MD17** [34] provides molecular dynamics trajectories of small organic molecules, with both energy and force labels. (4) **MPtrj** [35] includes 1.58 million structures obtained from DFT relaxation trajectories of over 146,000 materials in the Materials Project [36].

### 4.1 Results of pre-training and Fine-tuning

**Setup.** We pre-train our model on PCQM4Mv2 and subsequently fine-tune it exclusively on MD17. We adopt the Equiformer-V2 backbone [24]. Note, we use only the 3D structures without the property values in PCQM4Mv2. During the fine-tuning, given that the pre-trained AniDS has already learned a decent distribution for adding noise, we freeze the noise generator and only train the denoising autoencoder, using AniDS as an auxiliary task along with supervised learning. Following [27], we use 950 molecules for training and 50 for testing. No noise is added during validation and testing.

**MD17.** Table 1 presents the results on MD17. Our method achieves the best across all evaluated tasks. Compared to the previous results, we observe an average improvement of approximately **8.9%** across the tasks. We attribute this significant improvement to our structure-aware noise generator, which produces anisotropic noise, enabling our model to learn the molecular potential energy surface more effectively. Furthermore, while DeNS ($L_{max} = 3$) enhances input features by increasing the degree of irreducible representations in EquiformerV2, AniDS ($L_{max} = 2$) outperforms it while using a lower dimensional irreducible representation. This demonstrates the efficacy of our adaptive structure-driven noise sampling strategy, suggesting that well-informed noise learning can lead to a more generalizable force field prediction.

### 4.2 Results of Supervised Learning with Partial Corruption and Auxiliary Denoising

**Setup.** Here AniDS is used as an auxiliary task when conducting supervised learning on OC22. Due to resource constraints, we adopt the pre-trained models from [27]. This pre-trained model utilizes the EquiformerV2 architecture, and its training process incorporated DeNS as an auxiliary task. Given its architectural similarity to the model used for MD17, we can easily extend its training by integrating our AniDS framework. Following [27], we conduct training on OC22's S2EF-Total task, which contains 8.2 million structures. We further evaluate performance on the provided validation split [33]. Additionally, our results on MPTrj are provided in Appendix C.

Table 3: Ablation Studies on the Aspirin dataset of MD17. **(a)** Comparison of different denoising methods under the pre-train-fine-tune scheme. **(b)** Comparison of different denoising methods under the pre-train-fine-tune scheme. Standard supervised fine-tuning is used without AniDS as an auxiliary task. **(c)** Hyperparameter analysis under the pre-train and fine-tune scheme.

| (a) Pre-train and Fine-tune | | (b) *w/o* Auxiliary Fine-tuning | | (c) Hyperparameters | |
|---|---|---|---|---|---|
| **Model** | **F(MAE)** | **Model** | **F(MAE)** | **EqV2** | **F(MAE)** |
| – | – | – | – | $\sigma_p = 0.5$ | 0.13018 |
| EqV2-No pre-train | 0.1929 | EqV2-No pre-train | 0.1929 | $\sigma_p = 0.1$ | **0.11180** |
| EqV2-DeNS | 0.1178 | EqV2-DeNS | 0.1424 | $\sigma_p = 0.05$ | 0.11594 |
| EqV2-DenoiseVAE | 0.1143 | EqV2-DenoiseVAE | 0.1400 | $\lambda_{KL} = 1.5$ | 0.11714 |
| EqV2-AniDS-add | 0.1128 | EqV2-AniDS-add | 0.1389 | $\lambda_{KL} = 1.0$ | **0.11180** |
| **EqV2-AniDS** | **0.1118** | **EqV2-AniDS** | **0.1369** | $\lambda_{KL} = 0.5$ | 0.11461 |

**OC22.** Table 2 presents the results. Compared to baselines, AniDS achieves state-of-the-art performance across all tasks. This is because AniDS can more accurately generate noise that aligns with the atomic vibration modes, thereby better assisting the model to learn molecular force fields. Notably, our model demonstrates an improvement of ∼**3.2%** in energy prediction for both the in-distribution (ID) and out-of-distribution (OOD) datasets, and an improvement of ∼**6.2%** in force prediction for both ID and OOD datasets. These superior results underscore the effectiveness of our method.

## 4.3 Ablation Study and Hyperparameter Analysis

**Impact of Different Denoising Methods.** We evaluate various denoising strategies with the same Equiformer-v2 backbone [24], pre-trained on the PCQM4Mv2 dataset. The methods include: (1) no pre-training; (2) denoise training with fixed-scale isotropic Gaussian Noise (EqV2-DeNS [27]); (3) DenoiseVAE [16]; (4) AniDS with additive anisotropic correction instead of subtractive correction (*cf.* Equation 14; EqV2-AniDS-add); and (5) our method AniDS.

Table 3 compares two fine-tuning settings following denoising pre-training: (a) fine-tuning with denoising as an auxiliary task, and (b) fine-tuning without denoising. In both settings, our model consistently outperforms all baselines. All denoising-based pre-training methods substantially outperform the baseline without pre-training, supporting the benefit of learning molecular noise distributions. Notably, models using anisotropic noise (EqV2-AniDS and EqV2-AniDS-add) consistently outperform those using isotropic noise (EqV2-DeNS and EqV2-DenoiseVAE), confirming that isotropic assumptions oversimplify molecular dynamics. Moreover, the subtractive correction in EqV2-AniDS yields slightly better results than the additive correction in EqV2-AniDS-add, suggesting that incorporating directional stiffness from physicochemical priors improves the model's structural understanding.

**Impact of Different $\sigma_p$ and $\lambda_{KL}$.** The choice of prior distribution $\sigma_p$ and the strength of the KL loss $\lambda_{KL}$ both affect model performance. As shown in Table 3(c) (top), varying $\sigma_p$ influences how effectively the model learns from the noise prior, with the best performance achieved at $\sigma = 0.1$. In addition, Table 3c (bottom) shows the effect of different $\lambda_{KL}$ values. The KL loss helps prevent trivial solutions (*e.g.,* collapsing to zero noise) by regularizing the learned noise distribution. Our results indicate that setting $\lambda_{KL} = 1.0$ yields the best performance. Larger values overly bias the model toward the prior, ignoring structure-specific features, while smaller values lead to under-regularized, uninformative noise distributions. Together, these results highlight the importance of carefully balancing prior assumptions and regularization strength when modeling physically meaningful noise.

## 4.4 Case Study: Validating Anisotropic Noise Behavior on Crystal and Molecular Structures

To verify whether the learned noise captures the physical anisotropy of the energy landscape, we design an experiment to probe how directional perturbations influence energy changes. To quantitatively evaluate this effect, we first perform eigenvalue decomposition on the learned covariance matrices and apply small random perturbations along the resulting eigenvectors. The corresponding changes in energy are then measured using the Symmetric Mean Absolute Percentage Error (sMAPE).

**Crystal Validation.** The resulting energy changes, along with the associated eigenvectors and eigenvalues, are visualized in Figure 3(c).

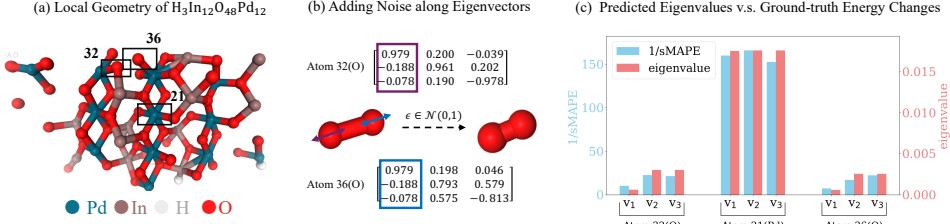

Figure 3: Visualization of the $H_3In_{12}O_{48}Pd_{12}$ crystal. We select oxygen atoms at indices $\{32, 36\}$, and a palladium atom at index 21. Figure (b) presents the eigenvectors of the oxygens. Figure (c) shows the relationship between the structural energy and applied noise.

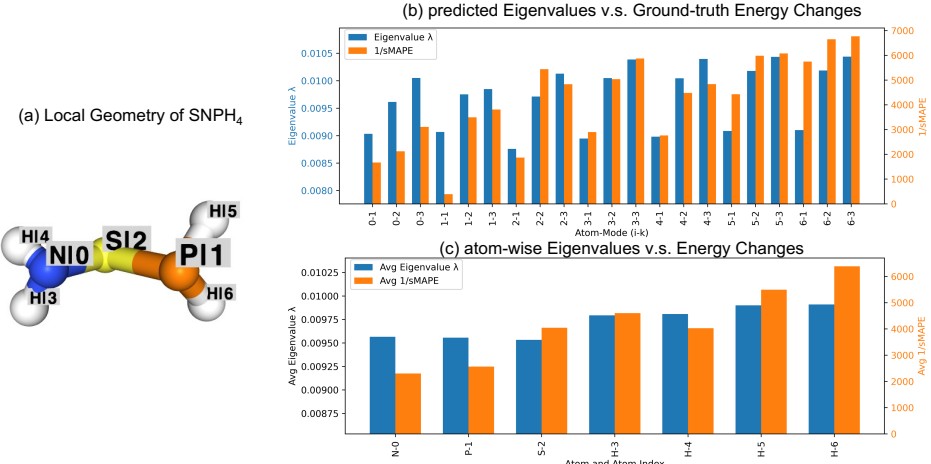

Figure 4: Analysis on the $SNPH_4$ molecule. (a) Local geometry showing atom indices. (b) Per-atom directional alignment between predicted eigenvalues and ground-truth energy sensitivities (1/sMAPE). (c) Atom-wise comparison of averaged eigenvalues and energy sensitivities.

For example, atoms $\{32, 36\}$ are spatially close, identical oxygen atoms with strong repulsive interaction. **The eigenvector corresponding to the smallest eigenvalue for both atoms aligns with the bond direction between them (Figure 3b).** Due to the short distance and strong repulsion, perturbations along this axis lead to significant energy changes, and AniDS accordingly reduces noise in this direction. In contrast, atom 21 resides in a more symmetric environment. The model thus produces larger and more isotropic noise for this atom. This strongly supports our main idea: AniDS allows for structure-aware noise generation, reducing excessive disruption.

**Molecular Validation.** To further confirm the generality of the anisotropic noise modeling, we conduct an additional analysis on the $SNPH_4$ molecule.

Figure 4(b) shows that the predicted eigenvalues exhibit a consistent trend with the ground-truth energy changes measured by 1/sMAPE, demonstrating that AniDS effectively captures direction-dependent energy stiffness. Notably, for atoms H6–1 and H5–1, the smallest eigenvalues correspond to eigenvectors aligned with the two P–H bonds, matching the physically stiff directions of the molecule. Figure 4(c) further summarizes the per-atom averaged behavior. Hydrogen atoms display larger eigenvalues and higher 1/sMAPE values, indicating that their displacements have relatively minor effects on total energy. In contrast, the N, P, and S atoms situated at the core of the molecular structure and show smaller eigenvalues, signifying greater energy sensitivity to positional perturbations. The results show that perturbations along different directions affect the potential energy surface to varying degrees, validating the rationale behind the anisotropic noise design: **AniDS suppresses perturbations in energy-sensitive directions while allowing more noise along flexible axes, closely aligning with the ground-truth energy changes.**

## 5   Related Works

**Coordinate Denoising for 3D Atomistic Systems.** Coordinate denoising has been widely explored for pretraining on 3D atomistic systems [12, 13, 31, 17, 22], showing strong performance in predicting

quantum chemical properties, force fields, and energies [44, 45, 18]. While the method was originally proposed to equilibrium molecular structures, it is later extended to non-equilibrium systems by incorporating atomic force information into the model [27]. Coordinate denoising's effectiveness stems from its theoretical connection to learning molecular force fields [13], under the assumption that the noise distribution is isotropic Gaussian and the data distribution is a mixture of such Gaussians. To overcome the limitations of the isotropic assumption, Frad [17] combines torsional and coordinate noise, while Slide [18] applies independent Gaussian noise on bond lengths, angles, and torsion angles. However, these methods use fixed noise scales, ignoring the variability of energy potentials across distinct structures. Addressing this, DenoiseVAE [16] leverages a Variational Autoencoder [46] to learn atom-wise adaptive noise variances. In contrast, AniDS learns a full covariance matrix for each atom's noise distribution, enabling the modeling of anisotropic and structure-aware noise.

**Other 3D Molecular Pretraining Methods.** A representative approach involves directly pretraining on large-scale molecular dynamics simulation datasets. For example, MatterSim [47] and JMP [48] are trained on large-scale molecular dynamics trajectories and achieve strong performance on downstream property prediction tasks. However, these methods rely heavily on high-quality large-scale simulation datasets, which are often expensive to obtain and limited in scalability. Another line of work focuses on integrating multiple structural representations to improve model expressiveness. For instance, MoleculeSDE [49] and Transformer-M [50] combine 2D topological graphs with 3D geometric point clouds, using complementary perspectives to enhance molecular representations. UniCorn [51] unifies three mainstream self-supervised strategies into a multi-view contrastive learning framework to construct more comprehensive molecular representations. MoleBlend [52] fuses 2D and 3D structural information at the atomic relation level, enabling fine-grained structural modeling. Recent advances further explore cross-modal and language-grounded representations for molecular systems. 3D-MoLM [53] and MolCA [54] align molecular graphs and textual descriptions through multimodal pretraining, while NEXT-MOL [55] and ReactXT [56] integrate 3D diffusion or reaction-context modeling with language understanding. SIMSGT [57] revisits tokenization and decoding for masked graph modeling, and ProtT3 [58] extends text-based pretraining to proteins. Beyond molecule-text alignment, UAE-3D [59] propose a unified latent space for 3D molecular diffusion, and scMMGPT [60] leverage language models for single-cell representation learning. At a broader scale, NatureLM [61] envisions a general language of natural sciences, and deep-learning frameworks such as [62] highlight the potential of AI to uncover physical principles in crystalline materials.

## 6 Conclusion and Future Work

We present AniDS, a denoising framework that learns anisotropic, structure-aware noise distributions to enhance molecular force field modeling. By generating atom-specific full covariance matrices conditioned on molecular geometry, AniDS lifts the isotropic and homoscedastic assumptions inherent in prior denoising methods. Grounded in theoretical connections to force field learning, AniDS supports both pretraining and auxiliary fine-tuning. Extensive experiments on MD17 and OC22 show that AniDS achieves leading performance. Looking forward, we aim to extend AniDS to larger and more complex systems, such as proteins and RNAs, and apply it for molecular simulations.

## 7 Acknowledgement

This work was supported in part by the Ministry of Education (MOE T1251RES2309 and MOE T2EP20125-0039) and the Agency for Science, Technology and Research (A*STAR H25J6a0034).

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
