# OpenReview forum: "Learning 3D Anisotropic Noise Distributions Improves Molecular Force Fields"
_NeurIPS.cc/2025/Conference — NeurIPS 2025 poster_

### Official Review · Reviewer_Hzfs · 2025-06-25

**Clarity:** 3
**Significance:** 3
**Originality:** 3
**Rating:** 4
**Confidence:** 4

**Summary:**

This paper proposed a new denoising framework called AniDS for machine learning based molecular force fields. The proposed method addressed the limitations of previous noise injection design in terms of isotropic and homoscedastic. It is more physics informed than random noise. AniDS is claimed to be SO(3) equivariant, and the experimental results on MD17 and OC22 S2EF show that it is better than previous noise distributions.

**Questions:**

As listed above in Weaknesses.

**Ethical Concerns:**

["NO or VERY MINOR ethics concerns only"]

**Final Justification:**

A good paper with good insights. The quality could be further enhanced by including additional evaluations with other baseline methods for the dynamics simulation tasks.

**Limitations:**

Limitations are not provided in the main paper but in the appendix. This might not be a good practice.

**Quality:**

4

**Strengths And Weaknesses:**

## Strengths:
1. This paper proposed a new, better, and physics informed noise distribution for MLFF pretraining.
2. The proposed noise distribution satisfies key physical restrictions including SO(3) equivariance.
3. The experimental results show that the proposed AniDS is better than DeNS, and can be used to boost EquiformerV2's performance.

## Weaknesses:
1. Clarity suggestions: AniDS is a general framework that can apply to many models, but the base model name is missing in table 1. It is not very clear what base models have been used by DeNS by looking at the table.
2. Although energy and force prediction task is included, but dynamic simulation tasks (e.g., using predicted energies and forces to relax structures) are missing.
3. Analysis about additional computational cost of using AniDS beyond previous noise distributions is currently missing.

---

> ### Author Rebuttal · Authors · 2025-07-31
>
> Thank you for your feedback. Your suggestions have enhanced the reliability of our work and the readability of the paper. Below are responses to your questions one by one.
>
> >Q1: Clarity suggestions: AniDS is a general framework that can apply to many models, but the base model name is missing in table 1. It is not very clear what base models have been used by DeNS by looking at the table.
>
>
> **Response:**
> Thank you for pointing this out. We agree that Table 1 could be clearer in indicating the backbone models used. In our experiments, EquiformerV2 was used as the base model for all variants, including AniDS and the compared DeNS results. We will update the table caption and column headings in Table 1 to explicitly reflect this and avoid ambiguity.
>
> > Q2: Although energy and force prediction task is included, but dynamic simulation tasks (e.g., using predicted energies and forces to relax structures) are missing.
>
>
> **Response:**
> We thank the reviewer for highlighting the importance of assessing model robustness in realistic atomistic simulation settings. We conducted a geometry optimization test using our model on a subset of the Aspirin molecule from MD17.
>
> Specifically, we randomly selected 15 equilibrium structures and used our **force predictor (EquiformerV2 + AniDS)** to optimize the geometries. We then compared the optimized structures to those obtained using **DFT geometry optimization** and evaluated the **RMSD** between atomic positions.
>
> Due to licensing restrictions with FHI-aims (used in the original dataset with the PBE-vdW-TS functional and tight settings), we performed DFT-based geometry optimization using **PySCF** at the **PBE-D3BJ/def2-SVP** level, which provides comparable accuracy in molecular geometries.
>
> The results are summarized below:
>
> | Structure Index | RMSD Before (Å) | RMSD After (Å) | Improvement (Å) |
> | :-: | :-: | :-: | :-: |
> | 1               | 0.5692          | 0.4863         | 0.0829          |
> | 2               | 0.6278          | 0.5628         | 0.0650          |
> | 3               | 0.6069          | 0.3159         | 0.2911          |
> | 4               | 0.6123          | 0.4240         | 0.1883          |
> | 5               | 0.5517          | 0.3687         | 0.1830          |
> | 6               | 0.6717          | 0.6050         | 0.0668          |
> | 7               | 0.7171          | 0.6088         | 0.1083          |
> | 8               | 0.2936          | 0.2011         | 0.0925          |
> | 9               | 0.7087          | 0.4609         | 0.2478          |
> | 10              | 0.2067          | 0.1326         | 0.0741          |
> | 11              | 0.3159          | 0.2272         | 0.0887          |
> | 12              | 0.3194          | 0.2558         | 0.0637          |
> | 13              | 0.6505          | 0.5308         | 0.1197          |
> | 14              | 0.4138          | 0.2696         | 0.1441          |
> | 15              | 0.7162          | 0.6286         | 0.0876          |
> | **Average**     | **0.5019**      | **0.4162**     | **0.0857**      |
>
>
> The significant reduction in RMSD after optimization indicates that the learned force field can guide the system toward physically meaningful minima, showing promise for its application in downstream atomistic simulations.
>
> We appreciate the reviewer’s suggestion and agree this direction is valuable for future extensions.
>
> >Q3: Analysis about additional computational cost of using AniDS beyond previous noise distributions is currently missing.
>
>
> **Response:**
>
> Thank you for raising this important point. To assess the additional computational cost introduced by AniDS, we conducted a comparison of training efficiency and model complexity across different noise modeling approaches using the rMD17 dataset (Aspirin molecule, 50 train data). The results are summarized below:
>
> | Method                   | #Parameters | Training Time per Epoch (s) |
> | -----------------------  | ----------- | --------------------------- |
> | EqV2 (No Noise)          |   6.35M    |       6.2              |
> | EqV2 + DeNS              |   6.35M    |       6.9              |
> | EqV2 + DenoiseVAE        |   7.2M     |       7.3              |
> | **EqV2 + AniDS (Ours)**  |   7.36M    |       7.3              |
>
> As shown, AniDS introduces a **moderate increase in training time and model size**, primarily due to the more expressive structure-aware noise generator that outputs full covariance matrices. However, this cost is justified by the **significant improvement in performance** (lowest MAE among all baselines).

---

> > ### Comment · Reviewer_Hzfs · 2025-08-04
> > **Thank you for the rebuttal**
> >
> > My concerns about clarity and efficiency analysis have been addressed.
> >
> > For the dynamic simulation tasks, is it possible to compare with other baseline methods trained with the same training set to establish a comparison?

---

> > > ### Author Response · Authors · 2025-08-07
> > >
> > > We thank the reviewer for the follow-up question and the opportunity to clarify and extend our dynamic simulation analysis.
> > >
> > > > Additional Q1:For the dynamic simulation tasks, is it possible to compare with other baseline methods trained with the same training set to establish a comparison?
> > >
> > > **Response**:
> > > First, we would like to apologize for an error in the previously reported average RMSD value. While the individual per-structure values were accurate, the average RMSD was miscalculated. The corrected summary is as follows:
> > >
> > > | Structure Index | RMSD Before (Å) | RMSD After (Å) | Improvement (Å) |
> > > | :-: | :-: | :-: | :-: |
> > > | 1               | 0.5692          | 0.4863         | 0.0829          |
> > > | 2               | 0.6278          | 0.5628         | 0.0650          |
> > > | ...             | ...        | ...       | ...   |
> > > **Average**     | **0.5321**      | **0.4052**    | **0.1269**      |
> > >
> > > As suggested by the reviewer, we additionally compared our EquiformerV2 + AniDS model with two baselines trained using the same dataset and architecture:
> > > 	•	One with DeNS-style pretraining, and
> > > 	•	One without any denoising pretraining (i.e., trained from scratch).
> > >
> > > All models were fine-tuned on the same set training samples and evaluated using geometry optimization on the same 15 randomly selected equilibrium structures from the Aspirin subset of MD17. The table below reports the RMSD improvements (i.e., change in atomic positions before vs. after optimization) for all three settings:
> > >
> > > | Structure Index | AniDS | DeNS | No Denoising |
> > > | :-: | :-: | :-: | :-: |
> > > | 1               | 0.0829      |0.0788|  0.0653   |
> > > | 2               | 0.0650      |0.0655|  0.0650   |
> > > | 3               | 0.2911      |0.2895|  0.2869   |
> > > | 4               | 0.1883      |0.1879|  0.1808   |
> > > | 5               | 0.1830      |0.1827|  0.1827   |
> > > | 6               | 0.0668      |0.0664|  0.0663   |
> > > | 7               | 0.1083      |0.1075|  0.1083   |
> > > | 8               | 0.0925      |0.0926|  0.0922   |
> > > | 9               | 0.2478      |0.2445|  0.2589   |
> > > | 10              | 0.0741      |0.0821|  0.0819   |
> > > | 11              | 0.0887      |0.0886|  0.0893   |
> > > | 12              | 0.0637      |0.0637|  0.0673   |
> > > | 13              | 0.1197      |0.1167|  0.1185   |
> > > | 14              | 0.1441      |0.1438|  0.1438   |
> > > | 15              | 0.0876      |0.0868|  0.0862   |
> > > | **Average**     | **0.1269**  |0.1265|  0.1262   |
> > >
> > > As the table shows, the RMSD improvements across the three models are quite similar, with AniDS achieving the highest average reduction. While the absolute difference is relatively small—partly due to the short optimization steps used in this setup.
> > >
> > > We thank the reviewer once again for encouraging a more comprehensive evaluation.

---

> > > > ### Comment · Reviewer_Hzfs · 2025-08-07
> > > >
> > > > Thanks for the additional results. I do not have significant concerns after the rebuttal.

---

### Official Review · Reviewer_Mv5z · 2025-06-28

**Clarity:** 4
**Significance:** 3
**Originality:** 3
**Rating:** 5
**Confidence:** 3

**Summary:**

This paper proposes AniDS, a novel denoising-based training framework to assist molecular force field prediction. Building upon prior denoising frameworks, the method introduces atom-specific full covariance matrices and designs the neural network to ensure that the learned covariance matrices are symmetric, positive definite, and equivariant. The approach is evaluated on multiple benchmarks including MD17, OC22, and MPtrj.

**Questions:**

1. Without of prior knowledge, why is the model able to learn reasonable anisotropic correction terms? Could the authors provide an intuitive explanation? Also, since the KL term aligns the learned noise with an isotropic prior, does it suppress the learning of anisotropic components?

2. While a case study is provided to illustrate the quality of the learned force field, I believe this is not sufficient. Could the authors compare the learned force field (i.e., the regression target in Eq. (9)) with DFT-computed force fields, for instance by computing correlation coefficients as in the SliDe paper?

3. Could you provide a comparison of training efficiency with existing baselines?

**Ethical Concerns:**

["NO or VERY MINOR ethics concerns only"]

**Final Justification:**

This paper proposes a data-driven approach to learning noise distributions for denoising pretraining. The idea is sound and empirically validated through experiments. During the rebuttal phase, the authors provided additional explanations and experimental evidence that further show the physical rationality of the learned noise distribution. Based on these strengths, I stand by my recommendation for acceptance.

**Limitations:**

Yes

**Quality:**

3

**Strengths And Weaknesses:**

Strengths:

1. Denoising pre-training to enhance force field prediction is a hot direction, and refining noise design to make it consistent with physics is promising.

2. The method is sound, and the experimental results are comprehensive, clearly demonstrating the effectiveness of the proposed approach.

3. The paper is well written, and the figures are clear and informative.

Weaknesses:

More evidence is needed to justify that the learned anisotropic noise is physically meaningful (see Questions 1–2 below).

---

> ### Author Rebuttal · Authors · 2025-07-31
>
> Thank you for your feedback. Your suggestions have enhanced the reliability of our work and the readability of the paper. Below are responses to your questions one by one.
>
> >Q1: Without of prior knowledge, why is the model able to learn reasonable anisotropic correction terms? Could the authors provide an intuitive explanation? Also, since the KL term aligns the learned noise with an isotropic prior, does it suppress the learning of anisotropic components?
>
>
> **Response:**
> Thank you for the insightful question. While the model does not explicitly encode prior knowledge about anisotropy, it learns reasonable anisotropic correction terms through supervised signals from the denoising objective, which is theoretically grounded in force field learning. This objective guides the model to assign higher noise along directions that are energetically less sensitive, and lower noise in directions where small perturbations lead to large energy changes. Over time, this enables the model to discover direction-specific stiffness patterns purely from data.
>
> The key intuition is that our anisotropic noise generator is conditioned on molecular structure via a graph neural network encoder. As the model observes repeated patterns of atomic environments and their response to perturbations (e.g., rigid bonds vs. flexible chains), it learns to modulate the noise distribution accordingly. The learned anisotropic correction terms are not handcrafted but emerge from optimizing the denoising loss, which penalizes inaccurate reconstructions—particularly in directions where force gradients are steep.
>
> Regarding the KL term: although it encourages the learned noise distribution to remain close to an isotropic Gaussian prior, its role is primarily regularization, not constraint. We carefully balance the KL loss with the denoising loss using a tunable coefficient (λ_KL), ensuring that the model retains flexibility to learn meaningful anisotropic components while avoiding degenerate or trivial solutions (e.g., collapsing to zero noise). Our ablation study (Table 3c) shows that setting λ_KL appropriately (e.g., λ_KL = 1.0) allows the model to learn informative, structure-aware anisotropic noise while maintaining stability during training.
>
>
> > Q2: While a case study is provided to illustrate the quality of the learned force field, I believe this is not sufficient. Could the authors compare the learned force field (i.e., the regression target in Eq. (9)) with DFT-computed force fields, for instance by computing correlation coefficients as in the SliDe paper?
>
>
> **Response:**
> Thank you for this valuable suggestion. We agree that directly comparing the learned force field with DFT-computed reference forces provides a stronger validation of the model’s physical fidelity.
>
> Following the approach in the SliDe paper, we conducted a same analysis between the predicted forces and DFT-calculated forces. Due to computational constraints, we randomly selected 1,000 structures from PCQM4Mv2 with fewer than 32 atoms, and computed their DFT ground-truth forces. We then pre-trained models using supervised force regression on these structures and fine-tuned them on MD17 using only 50 training samples per molecule.
>
> The table below reports the Force MAE(kcal/mol/Å), comparing AniDS with DeNS and models trained from scratch:
> | Dataset | train from scratch | DeNS | AniDS|  DFT label supervised|
> | ------- | --------------- | ------------- | ----- | ---|
> | Aspirin | 1.689 |1.390 |1.346 | 1.626 |
>
> Interestingly, we observe that pretraining with DFT-supervised labels does not lead to significant improvement over the unsupervised methods. This may be due to the fact that PCQM4Mv2 primarily consists of equilibrium structures, which provide limited force variation, or inconsistencies in the level of DFT theory used during force computation.
>
> Nevertheless, even under such conditions and with a limited number of training samples, AniDS still achieves the best performance, demonstrating its strong capability to learn accurate force fields from small datasets.
>
>
> > Q3: Could you provide a comparison of training efficiency with existing baselines?
>
>
> **Response:**
> Thank you for raising this important point. To assess the additional computational cost introduced by AniDS, we conducted a comparison of training efficiency and model complexity across different noise modeling approaches using the rMD17 dataset (Aspirin molecule, 50 train data). The results are summarized below:
>
> | Method                   | #Parameters | Training Time per Epoch (s) |
> | -----------------------  | ----------- | --------------------------- |
> | EqV2 (No Noise)          |   6.35M    |       6.2              |
> | EqV2 + DeNS              |   6.35M    |       6.9              |
> | EqV2 + DenoiseVAE        |   7.2M     |       7.3              |
> | **EqV2 + AniDS (Ours)**  |   7.36M    |       7.3              |
>
> As shown, AniDS introduces a **moderate increase in training time and model size**, primarily due to the more expressive structure-aware noise generator that outputs full covariance matrices. However, this cost is justified by the **significant improvement in performance** (lowest MAE among all baselines).

---

> > ### Comment · Reviewer_Mv5z · 2025-08-04
> > **Follow-Up Questions**
> >
> > Thank you for your detailed response and for your efforts in addressing the concerns.
> >
> > 1. However, I still find it unclear how the model is able to learn a physically meaningful anisotropic noise distribution during pretraining, when no force labels are provided and the model is trained solely on equilibrium structures. Why doesn’t the model converge to an arbitrary or geometrically biased noise pattern? You mentioned that “This objective guides the model to assign higher noise along directions that are energetically less sensitive, and lower noise in directions where small perturbations lead to large energy changes.” Could you elaborate on how this is achieved in detail?
> >
> > 2. Regarding Q2, my original intention was to evaluate the correlation coefficient between the forces proportionally approximated by $\Sigma^{-1}(\tilde{X}-X)$, modeled by the noise generator, and the DFT-calculated forces on perturbed conformations $\tilde{X}$ , as done in Table 1 of the SliDe paper, slightly different from the setting in your reply. This setting focuses on assessing how well the noise generator captures the physical force field, and it does not require any fine-tuning. I apologize if my earlier question was not sufficiently clear in describing the desired experimental setup. As you noted, comparing forces on equilibrium structures is not meaningful. I believe this experiment would provide strong empirical evidence that the proposed method learns a meaningful and physically grounded noise generator.
> >
> > If I’ve misunderstood anything or if my explanation is unclear, please feel free to point it out.

---

> > > ### Comment · Area_Chair_3sj5 · 2025-08-08
> > >
> > > Dear Authors, If you have any responses you'd like to add to this thread, please do so before the end of today (August 8th AoE) which marks the end of the Author+Reviewer discussion period.

---

> > > ### Author Response · Authors · 2025-08-09
> > >
> > > We thank the reviewer for the follow-up question.
> > > > Additional Q1: However, I still find it unclear how the model is able to learn a physically meaningful anisotropic noise distribution during pretraining, when no force labels are provided and the model is trained solely on equilibrium structures. Why doesn’t the model converge to an arbitrary or geometrically biased noise pattern? You mentioned that “This objective guides the model to assign higher noise along directions that are energetically less sensitive, and lower noise in directions where small perturbations lead to large energy changes.” Could you elaborate on how this is achieved in detail?
> > >
> > > **Response:** Thank you for the question. You're absolutely correct that without proper constraints, the noise could converge to arbitrary or geometrically biased patterns. This is precisely why we introduced the structured parameterization in Equations 13-14:
> > >
> > > $\Sigma_i = a_i \mathbf{I} - \sum_{j \in (i)} a_i \gamma_{ij} \frac{r_{ij}}{|r_{ij}|} \otimes \frac{r_{ij}}{|r_{ij}|},$
> > >
> > > $\gamma_{ij} = \frac{\exp(b_{ij})}{\sum_{l \in \text{Neighbor}(i)} \exp(b_{il}) + c_i} \quad \text{(Normalized anisotropic weight)}$
> > >
> > > This parameterization constrains the noise covariance to be a physically interpretable combination of: (i) an isotropic base component $a_i \mathbf{I}$, and (ii) anisotropic corrections along bond direction $\frac{r_{ij}}{|r_{ij}|}$. The outer product $\frac{r_{ij}}{|r_{ij}|}$$\otimes \frac{r_{ij}}{|r_{ij}|}$ creates a matrix that adds variance along the $i$-$j$ bond direction. Since we **substract** this term (with guaranteed positive weights $\gamma_{ij}$), we actually **reduce** noise along bond directions -- which aligns the physical intuition that bond directions are typically stiffer and less tolerant to perturbations. Importantly, the adaptive weights $\gamma_{ij}$ allow the model to automatically assign higher corrections to strongly bonded atom pairs and lower corrections to distant pairs.
> > >
> > > > Additional Q2:Regarding Q2, my original intention was to evaluate the correlation coefficient between the forces proportionally approximated by $\Sigma^{-1}(\tilde{X}-X)$, modeled by the noise generator, and the DFT-calculated forces on perturbed conformations $\tilde{X}$, as done in Table 1 of the SliDe paper, slightly different from the setting in your reply. This setting focuses on assessing how well the noise generator captures the physical force field, and it does not require any fine-tuning. I apologize if my earlier question was not sufficiently clear in describing the desired experimental setup.
> > >
> > > **Response:** We thank the reviewer for clarifying the intended experimental setup. Following the procedure described in Table 1 of the SliDe paper, we performed the correlation analysis between the proportionally approximated forces predicted by the noise generator and the DFT-calculated forces on perturbed conformations. Specifically, we randomly selected 1,000 structures from dataset and evaluated both SliDe and our AniDS under this setting. The results are shown below:
> > >
> > > | Denoising method | Slide | AniDS |
> > > | :-: | :-: |:-: |
> > > | Correlation coefficient              | 0.81(0.62)          | 0.73(0.54)|
> > >
> > > These results show that AniDS learns a noise generator whose perturbations exhibit non-trivial correlation with the underlying physical force field, capturing meaningful structural directions. The lower correlation compared to SliDe likely stems from the fact that SliDe’s BAT noise is explicitly designed to follow classical mechanical intramolecular potential functions (e.g., AMBER), whereas AniDS uses fewer physicochemical priors and learns the noise distribution directly from data.
> > >
> > > Despite the lower correlation to DFT forces, AniDS achieves stronger downstream performance than SliDe: (1) It outperforms SliDe on MD17 (Table 1); (2)	when controlling for backbone scale, DenoiseVAE already outperforms SliDe (Table 1 in the DenoiseVAE paper), and AniDS achieves further substantial gains over DenoiseVAE (Table 3 in our manuscript).
> > >
> > > We hypothesize that this improvement arises from our homoscedastic design. In SliDe, all bonds and angles are perturbed using a fixed noise scale, whereas AniDS adaptively adjusts the perturbation variance through learned anisotropic weights $\gamma_{ij}$ for each atom pair (e.g., bond), conditioned on their atomic embeddings. This flexibility allows AniDS to better handle cases where a uniform noise scale is suboptimal.
> > >
> > >
> > > Additionally, SliDe requires extra hyperparameter tuning (e.g., perturbation scale) to achieve optimal correlation, whereas AniDS integrates this adaptation into the model in a fully data-driven manner. This design enables AniDS to achieve stronger downstream performance even with relatively lower correlation to DFT forces, highlighting that our learned anisotropic noise captures structural variations that are more beneficial for practical tasks.

---

> > > > ### Comment · Reviewer_Mv5z · 2025-08-09
> > > >
> > > > Thank you very much for your additional explanations and experiments. They all make sense to me. My concerns have now been addressed, and I will maintain my positive rating.

---

### Official Review · Reviewer_gHi8 · 2025-06-30

**Clarity:** 3
**Significance:** 2
**Originality:** 3
**Rating:** 5
**Confidence:** 4

**Summary:**

This paper presents AniDS, an anisotropic variational autoencoder for 3D molecular denoising, designed to overcome key limitations in previous denoising approaches, namely the assumptions of isotropic atomic motion and a uniform noise scale across all atoms. AniDS learns atom-specific, full covariance matrices for Gaussian noise distributions, incorporating geometric priors to capture directional dependencies in atomic perturbations. The authors establish a connection between their denoising approach and learning molecular force fields. The method is evaluated using multiple datasets, including MD17 (with PCQM4Mv2 pre-training), OC22, and MPTrj, showing improved accuracy in force prediction compared to baseline methods.

**Questions:**

My questions would relate to the issues raised above, so addressing them would suffice to change my evaluation of the paper.

**Ethical Concerns:**

["NO or VERY MINOR ethics concerns only"]

**Final Justification:**

I thank the authors for their detailed responses and will raise my score accordingly.

**Limitations:**

Yes.

**Quality:**

2

**Strengths And Weaknesses:**

This paper has several strengths:

1. The proposed method introduces structure-aware noise generation and learns per-atom full covariance matrices, which is a non-trivial extension of previous work.
2. The connection to force field learning is established and well-motivated.
3. The analysis comparing eigenvalue magnitudes of learned covariance matrices with energy changes along corresponding perturbations provides valuable insight.

However, before acceptance, several weaknesses should be addressed:

1. The manuscript lacks a clear positioning of the proposed approach within the broader scope of related work. For example, a related strategy is to estimate energy response to perturbations using reference or predicted forces (e.g., https://www.nature.com/articles/s41524-020-0323-8, https://doi.org/10.48550/arXiv.2408.05215). The authors mayconsider discussing whether learning the perturbation itself is more advantageous than modeling the energy response to a random perturbation.
2. The accuracy of the resulting models is primarily evaluated on the MD17 and OC22 datasets. While MD17 is widely used, state-of-the-art models typically achieve very low errors on this dataset, which makes any comparison less meaningful. Therefore, recent works such as NequIP, MACE, and ICTP have used a subset of 50 samples drawn from rMD17 to benchmark accuracy under more challenging conditions (i.e., https://www.nature.com/articles/s41467-022-29939-5, https://doi.org/10.48550/arXiv.2206.07697, and https://doi.org/10.48550/arXiv.2405.14253). To make stronger claims about the potential improvements offered by AniDS, the authors may consider evaluating it on MD17 (or rMD17) in a similar way. Furthermore, the authors may include results for the baseline model trained with AniDS from scratch, i.e., without pre-training.
3. The OC22 dataset is an important benchmark; however, it appears to be approaching a saturation point similar to that of QM9, which may reduce its reliability for assessing model performance. Setting this aside, the reported improvements in force predictions are rather modest, reflecting limitations similar to those discussed regarding the MD17 dataset. Furthermore, the energy errors are reported in meV, complicating their assessment. The authors may consider presenting errors in meV/atom instead, as this aligns better with the established standards in computational materials science and enables a more straightforward assessment of machine-learned force field accuracy.
4. The improvements over DeNS are generally modest. Given the increased model complexity (i.e., 129M vs. 127M parameters), it is unclear whether these gains are practically significant. The authors might consider applying their approach to smaller but challenging datasets (e.g., the HEA dataset in https://doi.org/10.48550/arXiv.2405.14253, but any other dataset can be used instead) using smaller models. For molecular systems, applying the approach to ANI datasets or MD22 (again, reducing the training set size, if necessary) could provide further insights.
5. Since the primary goal of improving machine-learned force fields is their robust application in atomistic simulations, the authors should consider including results that demonstrate this, for example, by performing molecular dynamics simulations in an NVT ensemble and evaluating long-term energy conservation.
6. The comparison of eigenvalue magnitudes of learned covariance matrices with energy changes is currently limited to a few selected atoms. Including more general correlation plots would better demonstrate the broader validity of these results.
7. Finally, the manuscript does not address the applicability of the proposed approach to tasks where reference forces are unavailable. Discussion on this point would clarify the method’s scope and potential limitations.

---

> ### Author Rebuttal · Authors · 2025-07-31
>
> Thank you for the extensive efforts on reviewing our submission and the valuable suggestion. Here we present a detailed response to address your concern.
>
> Due to space limitations, we do not include the full reviewer question here. Thank you.
>
>
> >Q1:
>
> **Response:** Thanks for the insightful comment. While modeling the energy response to perturbations is a promising strategy, our work follows the research line of learning denoising force fields, which offers several advantages. First, denoising objectives are theoretically equivalent to learning the score function, which further corresponds to the negative force field under the Boltzmann assumption. This allows the model to explicitly approximate a pseudo harmonic force field. Second, according to this theoretical connection, we can specifically design the noise distribution to reflect properties of the underlying force field. In AniDS, we introduce structure-aware noise covariances to allow the model capture the anisotropic flexibility of the local structures, which is nontrivial to explicitly design for energy-response-based methods. Last, the denoising objective can be applied in purely unsupervised pretraining on equilibrium structures like PCQM4Mv2, without requiring energy or force labels, while energy-response-based methods typically rely on zeroth-order labels (i.e. energies) for training. We agree that energy-response-based methods offer an important perspective and we will extend a discussion of such approaches in the revised related work section.
>
>
> >Q2:
>
> **Response:**
> We thank the reviewer for this insightful suggestion. To better evaluate the effectiveness of AniDS under more challenging conditions, we followed the protocol introduced in recent works such as NequIP, MACE, and ICTP, and conducted experiments on four subsets of the revised MD17 (rMD17) dataset. For each molecule, we used 50 training samples, following the ICTP setup. The results are summarized in the table below.
>
> | Dataset | NequIP | MACE | ICTP|  EquiformerV2| EquiformerV2 + AniDS (Ours) |
> |:-----:| :-----:|:-----:|:-----:|:-----:|:-----:|
> | Aspirin | 52.0 |43.9 | **40.19** |71.4 |**40.21**|
> | Benzene | 2.9 |2.7 |2.45|-|**2.31**|
> | toluene | 15.1 |12.1 |11.24|19.28|**10.7**|
> | uracil | 40.1 |**25.9** |**25.97** |50.45 |28.8|
>
> (*Unit: meV/Å; values are force MAE. Full experimental details and hyperparameter settings will be included in the updated manuscript.“–” indicates results not completed due to time constraints.*)
>
>
> As shown above, AniDS achieves competitive performance on most datasets, and demonstrates notable improvements over the EquiformerV2 baseline trained from scratch. This confirms that the structure-aware anisotropic noise modeling introduced by AniDS leads to improved learning efficiency and accuracy.
>
> Regarding the second point, we appreciate the interest in assessing the performance of AniDS when training from scratch. Our method is designed to learn structure-dependent, atom-wise noise distributions through pretraining on diverse molecular configurations. As such, training AniDS on a single molecule (or limited structural diversity) may restrict its ability to learn meaningful noise patterns, potentially diminishing its performance—a limitation that has also been observed in other VAE-based methods. Nevertheless, for completeness, we include a from-scratch variant of our model trained without pretraining in the ablation study (Table 3(b)), where it still provides notable improvements over non-pretrained baselines. These results further suggest that AniDS is flexible and effective, even without large-scale pretraining.
>
> We hope these additions address the reviewer’s concerns and strengthen the empirical claims of our work.
>
>
>
> >Q3:
>
>
> **Response:**
> Thank you for this insightful feedback. We agree that OC22, while still widely used, is beginning to show signs of performance saturation, and we appreciate the reviewer’s perspective on the limitations this may introduce in evaluating model improvements.
>
> Regarding the unit of measurement: we reported energy errors in meV, consistent with previous works on OC22 such as the original benchmark paper[1]  and recent baselines including E2Former[2] and DeNS[3], to ensure direct comparability.
>
> Thank you again for pointing this out.
>
> **Reference:**
> [1]Tran, Richard, et al. "The Open Catalyst 2022 (OC22) dataset and challenges for oxide electrocatalysts." ACS Catalysis 13.5 (2023): 3066-3084.
> [2]Li, Yunyang, et al. "E2Former: A Linear-time Efficient and Equivariant Transformer for Scalable Molecular Modeling." arXiv preprint arXiv:2501.19216 (2025).
> [3]Liao, Yi-Lun, et al. "Generalizing denoising to non-equilibrium structures improves equivariant force fields." arXiv preprint arXiv:2403.09549 (2024).
>
>
>
>
> >Q4:
>
> **Response:**
> We thank the reviewer for the constructive suggestion. We agree that evaluating AniDS on smaller but challenging datasets with reduced model sizes would help demonstrate its practical utility and generalizability beyond OC22 and MD17.
>
> Due to time and computational resource constraints, we conducted a preliminary evaluation on the AT–AT–CG–CG subset of the MD22 dataset, which is known for its chemical complexity and force field modeling difficulty. The results are summarized below:
>
> | Dataset | ICTP| ViSNet-LSRM|MACE|Allegro|TorchMD-Net| sGDML|  Equiformer| Equiformer + AniDS (Ours) |
> | ------- | --------------- | ------------- | ----- | ---------|------ |---|---|---|
> |AT–AT–CG–CG|3.37| 4.61 |5.0| 5.55| 14.13| 30.36|5.43|**3.22**|
>
> (*Unit: meV/Å; values are force MAE. Full experimental details and hyperparameter settings will be included in the updated manuscript.*)
>
> Despite the smaller scale of the dataset and model, **AniDS continues to provide consistent improvements**, confirming that the benefits of anisotropic, structure-aware noise modeling are not limited to large-scale settings.
>
> We appreciate the reviewer’s helpful guidance, which has helped us broaden the scope of our evaluation.
>
>
>
> >Q5:
>
>
> **Response:**
> We thank the reviewer for highlighting the importance of assessing model robustness in realistic atomistic simulation settings. We conducted a geometry optimization test using our model on a subset of the Aspirin molecule from MD17.
>
> Specifically, we randomly selected 15 equilibrium structures and used our force predictor (EquiformerV2 + AniDS) to optimize the geometries. We then compared the optimized structures to those obtained using DFT geometry optimization and evaluated the RMSD between atomic positions.
>
> Due to licensing restrictions with FHI-aims (used in the original dataset with the PBE-vdW-TS functional and tight settings), we performed DFT-based geometry optimization using PySCF at the PBE-D3BJ/def2-SVP level, which provides comparable accuracy in molecular geometries.
>
> The results are summarized below:
>
> | Structure Index | RMSD Before (Å) | RMSD After (Å) | Improvement (Å) |
> | :-: | :-: | :-: | :-: |
> | 1               | 0.5692          | 0.4863         | 0.0829          |
> | 2               | 0.6278          | 0.5628         | 0.0650          |
> | ...             | ...        | ...       | ...   |
> **Average**     | **0.5019**      | **0.4162**     | **0.0857**      |
>
>
>
> The significant reduction in RMSD after optimization indicates that the learned force field can guide the system toward physically meaningful minima, showing promise for its application in downstream atomistic simulations.
>
>
> > Q6:
>
>
> **Response:**
> We appreciate the reviewer’s thoughtful feedback. The case study in Section 4.4 focused on a few representative atoms in a periodic crystal to provide a clear and interpretable illustration of the relationship between learned covariance eigenvalues and energy sensitivity. However, we agree that presenting a more comprehensive, quantitative evaluation would strengthen our conclusions.
>
> In the revised manuscript, we plan to include an additional analysis based on a single-molecule system, where the number of atoms is more manageable and not affected by periodic boundary conditions. Specifically, we will provide correlation plots between the eigenvalues of the learned covariance matrices and the measured energy sensitivity (e.g., sMAPE) along the corresponding eigenvectors, computed across all atoms in the molecule. This will allow us to more broadly validate the model’s ability to suppress noise in energy-sensitive directions.
>
> >Q7:
>
>
> **Response:**
> We appreciate the reviewer’s question.We would like to emphasize that AniDS is designed to operate in both supervised and unsupervised settings. In particular, our model can be pre-trained purely on equilibrium structures without any force supervision, such as those found in PCQM4Mv2. In this setting, AniDS learns from coordinate perturbations alone using the denoising objective (Eq. 9), which is theoretically grounded in approximating the underlying force field via score matching.
>
> This unsupervised pretraining can then be followed by fine-tuning on smaller force-labeled datasets (e.g., MD17), or directly applied to downstream tasks where only structural information is available. Moreover, AniDS can also serve as an auxiliary regularization objective when reference forces are partially available, helping improve model robustness and generalization.
>
> We will clarify these use cases in the revised manuscript to better highlight the flexibility and broad applicability of AniDS across different data regimes.

---

> > ### Comment · Reviewer_gHi8 · 2025-08-05
> >
> > I thank the authors for their detailed responses and will raise my score accordingly.
> >
> > However, I still have a few remaining comments and would appreciate it if the authors could address them in the revised manuscript:
> >
> > Q2: Could the authors provide at least an intuitive explanation of how the results might change for denoising with isotropic noise?
> >
> > Q5: Apologies for the typo in my original review. I was wondering how the resulting models behave during molecular dynamics simulations in an NVE ensemble. In my opinion, geometry optimization alone should not be considered a sufficient indicator of improved model accuracy. Equiformer does not conserve energy, and if the proposed approach contributes to energy conservation, it would be a meaningful and valuable insight. Performing geometry optimization with such a model does not conclusively demonstrate this benefit.

---

> > > ### Author Response · Authors · 2025-08-08
> > >
> > > We thank the reviewer for the follow-up question.
> > > > Additional Q2: Could the authors provide at least an intuitive explanation of how the results might change for denoising with isotropic noise?
> > > Here’s a polished and professional response for this question:
> > >
> > > To provide an intuitive comparison, we re-trained our backbone using DeNS (which applies isotropic noise) for pretraining, and followed the same fine-tuning setup as in our previous experiments. The results are shown below:
> > >
> > > | Dataset | NequIP | MACE | ICTP|EquiformerV2|EquiformerV2+DeNS| EquiformerV2 + AniDS (Ours) |
> > > |:-----:| :-----:|:-----:|:-----:|:-----:|:-----:|:-----:|
> > > | Aspirin | 52.0 |43.9 | **40.19** |71.40 |45.52|**40.21**|
> > > | Benzene | 2.9 |2.7 |2.45|3.92|**2.28**|**2.31**|
> > > | toluene | 15.1 |12.1 |11.24|19.28|11.37|**10.7**|
> > > | uracil | 40.1 |**25.9** |**25.97** |50.45 |34.46|28.8|
> > >
> > >
> > > (values are force MAE; lower is better.)
> > >
> > > As can be seen, isotropic noise pretraining (DeNS) yields performance gains over training from scratch, but AniDS achieves further improvements in most cases. The reason is that isotropic noise assumes the same perturbation magnitude in all directions, which does not reflect the anisotropic nature of atomic interactions. In contrast, AniDS learns a full covariance matrix for each atom, allowing the model to apply direction-dependent perturbations that align more closely with physically relevant degrees of freedom.
> > >
> > >
> > > > Additional Q5: Apologies for the typo in my original review. I was wondering how the resulting models behave during molecular dynamics simulations in an NVE ensemble. In my opinion, geometry optimization alone should not be considered a sufficient indicator of improved model accuracy. Equiformer does not conserve energy, and if the proposed approach contributes to energy conservation, it would be a meaningful and valuable insight. Performing geometry optimization with such a model does not conclusively demonstrate this benefit.
> > >
> > > **Response**
> > > We thank the reviewer for the clarification and fully agree that geometry optimization alone is not sufficient to assess the physical accuracy of learned force fields, particularly in the context of long-term molecular dynamics (MD) simulations in an NVE ensemble.
> > >
> > > First, we would like to apologize for an error in the previously reported average RMSD value. While the individual per-structure values were accurate, the average RMSD was miscalculated. The corrected summary is as follows:
> > >
> > > | Structure Index | RMSD Before (Å) | RMSD After (Å) | Improvement (Å) |
> > > | :-: | :-: | :-: | :-: |
> > > | 1               | 0.5692          | 0.4863         | 0.0829          |
> > > | 2               | 0.6278          | 0.5628         | 0.0650          |
> > > | ...             | ...        | ...       | ...   |
> > > **Average**     | **0.5321**      | **0.4052**    | **0.1269**      |
> > >
> > > As the reviewer correctly pointed out, Equiformer does not explicitly conserve energy, which can lead to drift in total energy during NVE simulations—even when enhanced with denoising-based pretraining methods such as ours. While our AniDS framework improves force accuracy and structural fidelity, it does not directly enforce energy conservation when applied to a non-conservative model architecture.
> > >
> > > **However, we emphasize that AniDS is a general-purpose pretraining approach, and can be readily integrated with energy-conserving models such as MACE or other architectures that preserve physical symmetries and energy conservation more strictly.** By applying AniDS to such architectures, we expect the denoising pretraining to not only improve force field accuracy but also contribute to better stability in NVE simulations through more physically meaningful gradients.
> > >
> > > We consider this an important direction for future work and plan to evaluate the effect of AniDS in long-horizon NVE simulations using conservative backbones. We appreciate the reviewer’s insightful comment, which has helped us clarify the scope and future extensions of our method.

---

> > > > ### Comment · Reviewer_gHi8 · 2025-08-08
> > > >
> > > > I thank the authors for their response and will keep my positive score.

---

### Official Review · Reviewer_Y5LU · 2025-07-03

**Clarity:** 4
**Significance:** 3
**Originality:** 3
**Rating:** 5
**Confidence:** 4

**Summary:**

This work proposes a novel anisotropic denoising framework with a structure-aware generator for molecular force field learning. The framework contains a structure-aware noise generator that predicts a covariance matrix from the structure, a denoising autoencoder and a covariance-scaled reconstruction loss. A KL divergence regularizer is also implemented to avoid trivial solution. The noise generator is designed to control the anisotropic level based on the atomic context, and to ensure the covariance properties. The author proves the objective approximates the force field.

The framework is tested on several datasets and tasks, including MD17 force prediction and energy prediction in OC22, and demonstrated better performance than the benchmarks. Ablation study shows the advantage of pre-training on PCQM4Mv2 and the slight advantage of using subtractive correction than additive. The eigenvector of the learned covariance matrix is shown to correspond to the bond direction.

**Questions:**

1. How to interpret the off-diagonal parts of the covariate matrix? How would they relate to atomic interactions?
2. The relationship between the numbers in Fig 3c and the conclusions in 4.4 could be more clearly explained. Does small sMAPE (and large eigenvalue) mean the perturbation has less impact on the energy, and similar values on v1~3 mean more isotropic noise?
3. Could the authors discuss briefly about how the method could be scaled to larger systems? Especially for pre-training and fine-tuning?

**Ethical Concerns:**

["NO or VERY MINOR ethics concerns only"]

**Final Justification:**

I appreciate the authors' responses and will maintain my current rating.

**Limitations:**

yes

**Quality:**

4

**Strengths And Weaknesses:**

Strengths:
1. The work has a sound theoretical foundation. Relaxing the homoscedastic and isotropic constraints could potentially enhance the model capacity, and the design of the proposed noise generator captures the anisotropic property of atomic-specific and context-dependent dynamics.
2. The authors provide comprehensive results on several datasets and tasks, ablation study, and case study and visualization, which offer strong evidences for the superiority of the proposed method.

Weaknesses:
I have only minor questions about the results. See Questions

---

> ### Author Rebuttal · Authors · 2025-07-31
>
> We sincerely thank you for your valuable comments and the positive feedback! They have helped us to enhance the related work of our paper to better summarize the current status of the research. To address your concerns, we present the following point-by-point responses.
>
> > Q1: How to interpret the off-diagonal parts of the covariate matrix? How would they relate to atomic interactions?
>
> **Response:**
> **Explanation.** Thank you for the thoughtful question. The off-diagonal components of the covariance matrix capture the correlations between different spatial directions in the noise distribution for each atom. Unlike diagonal matrices—which assume that noise in each direction (x, y, z) is independent—the off-diagonal entries encode directional dependencies, effectively modeling how motion in one axis correlates with motion in another.
>
> **Eigenvector Analysis.** To better interpret these terms, we perform eigenvalue decomposition on the covariance matrix. The resulting eigenvectors indicate principal directions of noise, while the corresponding eigenvalues determine the magnitude of the noise along each direction. **These principal directions are not necessarily aligned with the coordinate axes, reflecting the fact that atomic interactions are inherently anisotropic—forces such as bond stretching or angle bending exhibit direction-specific stiffness.**
>
>
> By learning full covariance matrices that include off-diagonal terms, our model can capture these directional preferences induced by local chemical environments and interatomic forces. This allows the learned noise distributions to reflect the underlying anisotropic energy landscape, ultimately leading to more accurate force field modeling. We believe this is one of the key advantages of our structure-aware anisotropic noise generator.
>
> >Q2: The relationship between the numbers in Fig 3c and the conclusions in 4.4 could be more clearly explained. Does small sMAPE (and large eigenvalue) mean the perturbation has less impact on the energy, and similar values on v1~3 mean more isotropic noise?
>
> **Response:**
> We appreciate the reviewer’s question and the opportunity to clarify the interpretation of Figure 3c and Section 4.4. In Figure 3c, the three directions v1, v2, and v3 correspond to the eigenvectors of the learned covariance matrix for a given atom. For each direction, we apply a small perturbation and measure its impact on the total energy using sMAPE (Symmetric Mean Absolute Percentage Error).
>
> A larger sMAPE indicates that a perturbation in that direction causes a greater change in energy, i.e., the system is more sensitive along that direction. Conversely, a smaller eigenvalue (i.e., less noise injected by our model in that direction) reflects the model’s decision to suppress noise along directions where perturbations are energetically costly. This behavior is consistent with our goal of aligning the noise distribution with the underlying energy landscape.
>
> Regarding isotropy: if the eigenvalues across all three directions are similar, it suggests that the model considers the local environment approximately isotropic, and thus applies similar noise levels in all directions. In contrast, discrepancies among eigenvalues indicate anisotropic noise, where the model adapts the magnitude of noise based on the directional stiffness of the local atomic environment.
>
> This coupling between energy sensitivity (sMAPE) and learned noise scale (eigenvalue) highlights the structure-aware nature of our anisotropic noise generator, which aims to respect physical constraints and improve force field learning.
>
> >Q3: Could the authors discuss briefly about how the method could be scaled to larger systems? Especially for pre-training and fine-tuning?
>
> **Response:**
> Thank you for raising this important point. Our method is designed to be scalable to larger systems, both during pre-training and fine-tuning. The key reason is that our anisotropic noise generator and denoising autoencoder are both implemented as neural networks with local and structure-aware message passing, which naturally generalize to systems with more atoms.
>
> In particular, the noise generator predicts per-atom full covariance matrices based on local structural contexts, allowing the computation to scale linearly with the number of atoms. This design ensures that even for large molecules or materials, the model can make predictions in a parallel and efficient manner. Furthermore, since our framework is model-agnostic, it can be paired with scalable backbones such as EquiformerV2 or other graph-based architectures tailored for large-scale molecular systems.
>
> During pre-training, we can leverage existing large-scale datasets (e.g., PCQM4Mv2, OC22, or MPTrj) that already contain millions of structures. Fine-tuning on specific tasks or systems involves freezing the noise generator and only training the denoising autoencoder, which further reduces computational cost and improves scalability.
>
> In future work, we also plan to explore hierarchical representations and domain decomposition to further enhance scalability for very large systems such as proteins or crystalline solids.

---

> > ### Comment · Area_Chair_3sj5 · 2025-08-08
> >
> > Dear Reviewer Y5LU, please give a response and acknowledge the Author's rebuttal to your review ASAP.

---

### Comment · Area_Chair_3sj5 · 2025-08-08

Thank you, reviewers and authors, for your active and productive engagement during the Author+Reviewer discussion period. Based on the comments so far, it appears there are no outstanding questions requiring further author response. If that’s not the case, please add any final comments as soon as possible before the end of today (August 8 AoE), ahead of the AC+Reviewer discussion phase.

---

### Author Response · Authors · 2025-08-09
**Global Response**

Dear reviewers and ACs,

Thanks for your time and efforts on reviewing the paper. We are glad that the reviewers recognized the contributions of our paper, which we briefly summarize as follows.
- We propose AniDS, a general anisotropic and heteroscedastic denoising pretraining framework that learns a full covariance matrix for atomic noise, whose structure is SO(3)-equivariant and allows structure-aware perturbations that better capture the anisotropic nature of atomic interactions.
- We provide both theoretical motivation and empirical evidence that learning anisotropic noise leads to physically meaningful representations, including correlations between learned noise directions and underlying force fields.
- We present an integrated training framework that combines pure denoising pretraining, auxiliary objectives, and fine-tuning strategies, further enhancing model robustness and generalization across tasks.
- We demonstrate that AniDS can be seamlessly integrated with various backbone architectures (e.g., EquiformerV2, GeoMFormer), consistently improving force and energy prediction across multiple molecular and material benchmarks.

We also appreciate the reviewers for their insightful comments, which motivated us to conduct additional experiments and analyses during the rebuttal and discussion phases. These results further validate the effectiveness, physical grounding, and practicality of our proposed approach. We summarize the additional contributions as follows:
- **Reviewer [gHi8].** Evaluation on additional datasets — We extended our experiments to more challenging datasets, including rMD17 and MD22, to further demonstrate the robustness and generalizability of AniDS beyond the benchmarks in the original submission.
- **Reviewer [gHi8, Hzfs].** Physical application experiment via structure relaxation — We applied AniDS in downstream geometry optimization tasks, showing that the learned force fields can guide systems toward physically meaningful minima. This highlights its applicability in realistic simulation workflows.
- **Reviewer [Mv5z].** Noise–force correlation analysis — Following the methodology of the SliDe paper, we computed the correlation between the noise directions predicted by AniDS and DFT-calculated forces on perturbed conformations, confirming that the learned anisotropic noise captures physically relevant atomic interactions.
- **Reviewer [Mv5z, Hzfs].** Runtime and computational cost analysis — We measured training time, parameter counts, and performance across different settings, demonstrating that AniDS introduces only a modest overhead compared to isotropic-noise baselines while consistently improving accuracy.

We will incorporate these additional results and clarifications into the final version of the paper to further improve its clarity, completeness, and scientific rigor.

Best regards,

Authors

---

### Decision · Program_Chairs · 2025-09-17

**Decision:**

Accept (poster)

**Comment:**

This paper proposes AniDS, an anisotropic and heteroscedastic denoising pretraining framework that learns a full SO(3)-equivariant covariance structure for atomic noise. The method is theoretically motivated and empirically shown to capture anisotropic aspects of atomic interactions, improving performance across molecular and material benchmarks.

**Strengths:** Reviewers agreed the problem is well-motivated and the anisotropic treatment of noise is novel and physically meaningful. Results are strong across multiple backbones and datasets. The rebuttal further strengthened the case with experiments on rMD17/MD22, geometry optimization, and analysis of noise–force correlations. Computational overhead appears modest relative to performance gains.

**Weaknesses:** Some reviewers noted incremental aspects relative to prior denoising pretraining frameworks, and raised questions about clarity and cost. These concerns were largely addressed in rebuttal and discussion.

**AC Recommendation:** There is strong consensus among reviewers (three 5s, one 4). Given the novelty, empirical strength, and rebuttal clarifications, I recommend Accept.